# Advances in Ophthalmic Epigenetics and Implications for Epigenetic Therapies: A Review

**DOI:** 10.3390/genes14020417

**Published:** 2023-02-05

**Authors:** Spencer M. Moore, John B. Christoforidis

**Affiliations:** 1Department of Ophthalmology & Vision Science, University of Arizona College of Medicine-Tucson, Tucson, AZ 85711, USA; 2Retina Specialists of Southern Arizonam, Tucson, AZ 85712, USA

**Keywords:** epigenetics, retinal aging, retinal degeneration

## Abstract

The epigenome represents a vast molecular apparatus that writes, reads, and erases chemical modifications to the DNA and histone code without changing the DNA base-pair sequence itself. Recent advances in molecular sequencing technology have revealed that epigenetic chromatin marks directly mediate critical events in retinal development, aging, and degeneration. Epigenetic signaling regulates retinal progenitor (RPC) cell cycle exit during retinal laminar development, giving rise to retinal ganglion cells (RGCs), amacrine cells, horizontal cells, bipolar cells, photoreceptors, and Müller glia. Age-related epigenetic changes such as DNA methylation in the retina and optic nerve are accelerated in pathogenic conditions such as glaucoma and macular degeneration, but reversing these epigenetic marks may represent a novel therapeutic target. Epigenetic writers also integrate environmental signals such as hypoxia, inflammation, and hyperglycemia in complex retinal conditions such as diabetic retinopathy (DR) and choroidal neovascularization (CNV). Histone deacetylase (HDAC) inhibitors protect against apoptosis and photoreceptor degeneration in animal models of retinitis pigmentosa (RP). The epigenome represents an intriguing therapeutic target for age-, genetic-, and neovascular-related retinal diseases, though more work is needed before advancement to clinical trials.

## 1. Introduction

Epigenetic regulation of gene expression occurs via a wide spectrum of molecular chromatin modifications and may serve as an interface between environmental influences and transcriptional regulation. Epigenetic marks are dynamic molecular changes to the genome that regulate gene expression without changing the DNA base-pair sequence. They include DNA methylation and histone post-translational modifications—methylation, acetylation, ubiquitylation, and sumoylation. The epigenetic machinery encompasses hundreds of enzymes that may be categorized as writers (creating new epigenetic marks), readers (recognizing epigenetic marks), or erasers (removing epigenetic marks) [1]. Following the discovery of an evolutionarily conserved catalytic domain, the SET domain, common to a family of histone methyltransferases [2,3], there has been an explosion of work identifying additional families of epigenetic writers, readers, and erasers [4]. The writers include histone methyltransferases, histone acetyltransferases, and DNA methyltransferases. Readers include methyl-CpG-binding domain, Tudor domain, and bromodomain proteins. Erasers include ten-eleven translocation (TET) DNA demethylases, histone deacetylases (HDACs), and histone demethylases (HDMs) [4]. This diverse array of epigenetic machinery mediates numerous functions in the cell, ranging from development, the integration of environmental signals, and aging. The accessibility and well-defined laminar architecture of the mammalian retina has enabled the study of the retinal epigenome, with exciting recent developments in retinal aging. Epigenetic enzymes are also an emerging therapeutic target in numerous human diseases, including prospects for retinal pathology.

Epigenetic marks mediate the temporally and spatially dependent gene expression patterns that regulate development and tissue specificity. During fertilization, DNA is demethylated, effectively erasing the DNA methylome component of the epigenetic code [5,6]. In somatic cells during development, DNA methylation of pluripotency-associated genes occurs, whereas histone H3 lysine 27 (H3K27) is demethylated [5]. In the retina, development proceeds in a carefully orchestrated sequence in which retinal progenitor cells (RPCs) give rise to retinal ganglion cells (RGCs), horizontal cells, cones, and amacrine cells, followed later by rods, bipolar cells, and Müller glia [7]. H3K4me1 and H3K27ac, epigenetic marks of active transcription, regulate gene expression in RPCs, whereas these marks accumulate on photoreceptor-specific gene promoters later in development and in the mature retina [8]. Thus, the epigenome regulates retinal gene expression in a spatial and temporal manner which gives rise to the evolutionarily conserved laminar retinal architecture.

The accumulation of epigenetic marks across the genome, particularly DNA methylation at CpG (5′-cytosine-guanine-3′) islands, has emerged as a marker of and potentially a mechanism for molecular aging. Steve Horvath was an early pioneer of the “epigenetic clock” as a tool to estimate biological age in human tissues, using publicly available sequencing data to demonstrate a correlation between chronological age and CpG methylation data [9]. Recent work has shown that the epigenetic clock is also applicable to the retina, and intriguingly, some aspects of epigenetic aging may be reversible, with potential applications to glaucoma [10]. In this manuscript, we review recent advances in epigenetics in retinal biology by focusing on the epigenetic regulation of retinogenesis; epigenetic influences in RGC survival and degeneration; and the epigenetic molecular biology of photoreceptor development, homeostasis, and degeneration, as well as examining how the retinal epigenome serves as an integrator of environment signals. We also include discussion of future directions, including applications of epigenetic therapies to common retinal diseases, with selected findings summarized in Table 1.

## 2. Main Text

### 2.1. Epigenetic Regulation of Retinal Development

The evolutionarily conserved laminar organization and well-defined cell types of the mammalian neurosensory retina and retinal pigmented epithelium (RPE) enable detailed studies of the genetic and epigenetic mechanisms regulating development. The six neuronal (retinal ganglion cell (RGC), bipolar cell, amacrine cell, horizontal cell, rod photoreceptor, and cone photoreceptor) and one glial (Müller) cell types comprise the cellular structure of the neurosensory retina. Early neurogenesis and the birth of retinal progenitor cells (RPCs) is driven by the expression of retina transcription factors, including *PAX6* and *OTX2* [17]. Multipotent RPCs derived from neuroectoderm successively exit the cell cycle and give rise to an early population consisting of RGCs, cone photoreceptors, horizontal cells, and amacrine cells, followed by a late population consisting of rod photoreceptors, bipolar cells, and Müller glia [7,8,17]. Advances in next-generation sequencing technology have enabled detailed analysis of the transcriptional landscape of human fetal retinal development. Mellough et al. published a transcriptional characterization of the human fetal retina, describing three distinct developmental windows of retinal development [18]. Their first distinct developmental window, occurring between weeks 4–7 post-conception, included the expression of genes driven by the fibroblast growth factor (FGF) WNT, transforming growth factor-β/bone morphogenic protein (TGF-β/BMP), and sonic hedgehog (SHH) families [18]. In the second developmental window from weeks 8–10 post-conception, genes critical for RGCs, horizontal cells, amacrine cells, and cone photoreceptors were significantly upregulated [18]. In the third developmental window from weeks 12–18 post-conception, there was a switch from gene markers involved in proliferation to the expression of genes important for the differentiation of photoreceptors and phototransduction [18]. Various models of RPC differentiation into retinal neurons and glia have been described, including stepwise differentiation based on intrinsic transcriptional and epigenetic cues versus the reliance on extrinsic cues from the local environment [7]. 

Epigenetic regulation, including histone post-translational modification and DNA methylation, of retinal neurogenesis has been increasingly appreciated in recent work. Dynamic regulation of histone modification in the developing retina has been demonstrated spatially on both globally (i.e., physically across the layers of the retina) and locally (i.e., across the genome), as well as temporally throughout development. For example, H3K27me3 expression strongly colocalizes with the inner neuroblastic layer in the embryonic mouse retina, and in the inner nuclear and ganglion cell layers in the postnatal and adult mouse retina [19]. H3K9me2 was expressed in the embryonic mouse retina but decreased dramatically in the postnatal period [19]. In cells of the mouse photoreceptor lineage, H3K4me3 was strongly enriched, which correlated with the expression of photoreceptor-specific genes, whereas H3K27me3 was found to be enriched in bipolar lineage cells [20]. Moreover, amacrine and RGC-specific genes were suppressed by H3K27me3 expression [20]. Combined whole-genome bisulfite sequencing and RNA sequencing demonstrated an inverse correlation between DNA methylation and gene expression during retinal neurogenesis [21]. Numerous specific RPC genes have been shown to regulate development via epigenetic mechanisms. In the mouse embryonic retina, the retinal progenitor gene *Uhrf1* was enriched with the pro-transcriptional epigenetic marks H3K41/2/3, H3K27Ac, and H3K9/14Ac on its promoter during early development [21]. Interestingly, the related gene product *Uhrf2*, which is also important in RPC cell cycle regulation, aids in DNA demethylation, and *Uhrf2*^−/−^ mice overproduce RGCs through an overproliferation of RPC to RGC [22]. The H3K27me3 transcriptional repressor mark accumulated on the promoter and gene body of the basic helix-loop-helix (bHLH) RPC gene *Ascl1* as fate determination progressed and neurogenesis was complete [21]. As rod development progressed in the early postnatal mouse, dynamic epigenetic regulation of rod synaptic gene *Sitx3* was observed. Although H3K27me3 decreased at the promoter site over time, RNA PolII increased at the promoter and DNA methylation of the gene body decreased, which is consistent with the increased expression of *Sitx3* over this time period [21]. Moreover, changes in epigenetic marks and chromatin state were correlated with the developmental time-dependent expression of retinol-binding protein 3 (*RBP3/Rbp3*), a gene expressed in human and mouse rod photoreceptors [21]. Interestingly, epigenetic marks were found to colocalize in specific chromatin states within rod photoreceptors; for example, epigenetic marks associated with loose euchromatin such as H3K27Ac physically colocalized with euchromatin in the nuclear periphery, whereas epigenetic marks associated with heterochromatin such as H3K9me3 and H3K27me3 physically colocalized with more densely packed heterochromatin in the nuclear center [21]. Evidence suggests that, in the adult retina, epigenetic modifications directly regulate photoreceptor gene expression in a spatial manner; for example, outer nuclear layer (i.e., photoreceptor) cells demonstrate significantly reduced DNA methylation of photoreceptor genes *Rbp3* and *Rho* versus non-photoreceptor inner nuclear layer cells [23]. Thus, the authors concluded that DNA methylation functions to maintain cell-specific gene expression profiles in photoreceptors [23].

### 2.2. Epigenetic Basis of Aging and Disease in RGCs

Epigenetic chromatin modifications may represent a mechanism of molecular aging. Healthy aging is associated with epigenetic changes in multiple tissue types, including global decreases in DNA methylation and decreased H3K9me3 [24]. DNA methylation at CpG (5′-cytosine-guanine-3′) sites has been well validated as a molecular signature of aging [9]. Using thousands of human tissue samples with DNA methylation data, Steve Horvath developed a publicly available tool to calculate the age of a tissue sample based on its epigenetic signature, specifically based on CpG methylation [9]. Thus, Horvath’s “epigenetic clock” may be used to compare signs of epigenetic aging in the experimental setting [25]. A similar epigenetic clock is available for mouse tissues [26]. One interesting validation of the epigenetic clock applied to the retina revealed decelerated epigenetic aging in the retinas of *n* = 8 mice that spent 37 days in spaceflight aboard the International Space Station compared to *n* = 9 mice that remained on earth [27]. Indeed, the fetal retina has been used to demonstrate further validation of the epigenetic clock. Hoshino and colleagues found that in the human fetal retina, there was a strong correlation between chronologic age and epigenetic age, and they were able to track the transcriptional activation of photoreceptor genes during development [11]. Rather than with RPCs, retinogenesis began with RGCs and horizontal cells, followed by amacrine cells, rod photoreceptors, bipolar cells, and Müller glia [11]. They also found that epigenetic aging proceeded at a similar pace when retina explants were maintained in vitro, suggesting that epigenetic aging is an inherent property of the tissue [11]. Interestingly, the epigenetic clock demonstrated accelerated aging in the retina from trisomy 21 fetal retina explants versus controls [11].

#### Glaucoma and Dysregulated Epigenetic Aging 

Insights from epigenetics have shed light on the distinctions between normal aging and age-related retinal and optic nerve pathology such as glaucoma. Glaucoma encompasses a family of optic nerve disorders marked by the progressive irreversible degeneration of RGCs, which constitute the axons of the optic nerve. Age, family history, and intraocular pressure are strong risk factors, and glaucomatous optic neuropathies have been frequently used as experimental models for age-related neurodegeneration. Indeed, in a mouse RGC system, ectopic expression of three genes that are commonly used experimentally to induce pluripotency in vitro, *Klf4, Oct4,* and *Sox2*, also called Yamanaka factors after the landmark 2007 manuscript [28], altered DNA methylation profiles among aged mouse RGCs [10]. Notably, the fourth Yamanaka factor *Myc* was not included due to reduced mouse lifespan [10]. Lu and colleagues ectopically expressed *Klf4, Oct4*, and *Sox2* in mouse RGCs via an adenoviral (AAV2) vector via intravitreal injection. The expression of these factors reversed the DNA methylation changes associated with aging in RGCs that were accelerated by experimental optic nerve crush injury; moreover, the expression of these genes enhanced axonal regeneration following optic nerve crush [10]. They also found that for genes of which the expression differed between old and young mouse RGCs, Yamanaka factor expression restored a young-like transcriptomic profile in old mouse RGCs [10].

Other studies have investigated the recovery of RGCs following injury by manipulating the epigenome. In an NMDA-intravitreal injection model of RGC apoptosis, co-administration of an inhibitor of the bromo- and extraterminal domain reader of histone acetylation (BET), was protective against RGC cell death, although a molecular epigenetic mechanism was not proven in this study [29]. In a similar intravitreal NMDA injection model of RGC apoptosis, inhibition of transcription of the gene encoding histone lysine methyltransferase Ezh2 was protective against NMDA-induced apoptosis in RGCs by decreasing H3K27 trimethylation [30]. In another mouse optic nerve crush model, an assay for transposase-accessible chromatin sequencing (ATAC-seq) of sorted RGCs revealed differences in chromatin accessibility following optic nerve crushing [31]. Among genes located in regions of increased accessibility following injury were genes involved in endoplasmic reticulum stress and apoptosis, whereas genes involved in synaptic transmission and cell adhesion were found in areas of decreased chromatin accessibility, demonstrating the epigenetic underpinnings of the response to injury [31]. 

Early preclinical evidence has demonstrated epigenetic changes in glaucoma models similar to those seen in normal aging. Xu and colleagues harvested retinas from young (3 month) and aged (18 month) mice and performed ATAC-Seq and found significant differences in chromatin accessibility associated with aging, suggestive of possible underlying epigenetic mechanisms regulating the chromatin architecture with aging [12]. When young and aged mouse eyes were experimentally subjected to elevated intraocular pressure (IOP) in a glaucoma model, ATAC-Seq revealed increased chromatin accessibility in intronic areas of the genome associated with stress among aged retinas versus young retinas. Importantly, H3K27Ac, an active transcriptional marker, was increased in genes upregulated in the aged retinas in response to IOP elevation and in the young retinas in response to consecutive insults of IOP elevation [12]. Following IOP elevation, the H3K27Ac mark was enriched in genes associated with tumor necrosis factor-α (TNF-α) signaling, senescence, and age-related sterile inflammation (or so-called “inflammaging”) [12]. Finally, via DNA methylation analysis, the authors found that repeated IOP stress accelerated the epigenetic clocks in young retinas comparable to the aged retinas [12]. These results suggest the epigenetic control of transcription of aging-related genes via marks such as DNA methylation and H3K27 acetylation in response to environmental stressors such as a glaucomatous insult.

### 2.3. Epigenetic Mechanisms in Neovascular Retinal Conditions

#### 2.3.1. Epigenetic Basis of Age-Related Macular Degeneration

Age-related macular degeneration (AMD) is a complex blinding disease thought to originate in the retinal pigmented epithelium (RPE) and which has a complex environmental and genetic etiology, including potential epigenetic mechanisms. One early study showed that among monozygotic twin pairs, the twin with the more advanced AMD had a longer smoking history [32]. There were also inverse associations between AMD and dietary betaine intake and between drusen area and dietary methionine intake, which the authors postulated could have an epigenetic mechanism [32]. Peripheral blood from neovascular AMD patients exhibited hypomethylation of age-related maculopathy susceptibility locus 2 (*ARMS2*) [33]. Postmortem retinas from AMD patients showed hypermethylation of protease serine 50 *PRSS50* promoter [33]. The epigenetic clock as applied to AMD, however, has shown conflicting results. In one study, epigenetic clock data did not correlate significantly with chronologic age in blood or donor RPE in AMD patients [34]. However, the opposite was found in another, larger study. An epigenetic clock model revealed different aging signatures among gene expression profiles from four groups of postmortem retinas from AMD patients, varying in disease severity according to the Minnesota Grading System, showing that the epigenetic aging signature in AMD may indeed correlate with disease severity [35]. When DNA methylation analysis was performed on AMD and control postmortem human RPE cells, 3718 differentially methylated genes were found [36]. When these data were combined with gene expression data, 15 genes were hypermethylated and downregulated (i.e., epigenetically suppressed), and 19 genes were hypomethylated and upregulated (i.e., epigenetically induced) in AMD samples [36]. A CpG methylation model was also found to be predictive of AMD versus control samples. The authors also observed the hypomethylation of *SMAD2*, a gene involved in TGF-β signaling, which has been implicated in choroidal neovascularization (CNV) in AMD [36].

#### 2.3.2. Epigenetic Regulation of VEGF Expression

Vascular endothelial growth factor (VEGF) contributes to angiogenesis and CNV in AMD. In the oxygen-induced retinopathy mouse model of AMD, hypoxia leads to demethylation of the hypoxia-inducible factor-1α (HIF-1α) binding site and increased HIF-1α binding to the *Vegf* promoter [37]. The authors found that the knockout of aquaporin-4 (AQP-4) decreased hypoxia-induced hypomethylation of the *Vegf* promoter, thus postulating an environment–gene expression connection in which epigenetic regulators integrate hypoxic signals to influence gene expression in the retina [37]. In another mouse model of CNV, treatment with an HDAC inhibitor decreased the CNV area and vascular leakage in the retina [38]. The authors showed that the HDAC inhibitor increased acetylated H3 levels and decreased VEGF levels in the retina [38]. Dysregulated inflammatory signaling, including the recruitment of immune cells and pro-inflammatory cytokine signaling, plays a role in the multiple-hit model of AMD pathogenesis and may contribute to angiogenesis and CNV [39,40]. In monozygotic and dizygotic twin pairs which were discordant for AMD, peripheral blood mononuclear cells exhibited hypomethylation of CpG sites in *IL17RC* (a gene encoding a receptor for IL-17) and *CCL22* (a gene encoding a chemokine involved in T cell immunity) in the AMD subjects [41]. Moreover, *IL17RC* gene expression and IL17RC receptor expression was significantly elevated in postmortem AMD maculae versus non-AMD controls, although DNA methylation data were not available from these postmortem tissues [41]. Thus, the epigenome may integrate environmental signals such as hypoxia and inflammation to influence gene expression changes in AMD pathogenesis.

#### 2.3.3. Epigenetic Signals in Hyperglycemia and Diabetic Retinopathy

Diabetic retinopathy (DR) represents a common microvascular retinal complication of diabetes mellitus. Prolonged hyperglycemia may result in proliferative DR, in which neovascularization and tractional fibrotic membranes may result in irreversible vision loss. In advanced stages of DR, angiogenic signaling from the ischemic retina is mediated by VEGF, with emerging evidence for epigenetic mediators. Whole-blood DNA showed numerous differentially methylated CpG sites between DR and non-DR control patients [42]. Sirt6 is an HDAC, and deficiency is known to cause synaptic transmission deficits and apoptosis in the mouse retina [13]. In a non-obese diabetic (NOD) mouse model, retinas from diabetic mice showed decreased Sirt6 and increased acetylation of H3K9 and H3K56 versus controls. Diabetic mouse retinas also exhibited increased VEGF expression [13]. In addition, the authors demonstrated that Müller glia may be a source of Sirt6-mediated control of VEGF expression in response to high glucose: cultured Müller glia upregulated VEGF expression in response to short interfering RNA (siRNA) knockdown of *Sirt6,* and hyperglycemia-induced VEGF upregulation was blocked by *Sirt6* overexpression [13]. Thus, Sirt6 may represent an epigenetic environmental integrator in DR—downregulated *Sirt6* expression in response to hyperglycemia leads to decreased H3K56 deacetylation and upregulation of VEGF production in Müller glia [13]. Maternally expressed gene-3 (*Meg3*) encodes a long non-coding RNA (lncRNA) that is downregulated in a rat model of DR [14]. He and colleagues showed that the *Meg3* promoter is methylated by DNA methyltransferase 1 (DNMT1) [14]. Hyperglycemia in cultured rat endothelium led to the upregulation of DNMT1 expression, the suppression of *Meg3*, and the expression of proteins consistent with the endothelial-mesenchymal transmission and the activation of the mammalian target of rapamycin (mTOR) pathway [14]. The regulation of extracellular matrix protein expression by epigenetic modifications may also play a role in DR. Matrix metalloproteinase-9 (MMP-9) activity is increased in DR [43]. H3K9 methylation near the *Mmp-9* promoter decreased and H3K9 acetylation increased in a diabetic rat model, driven by increased lysine-specific demethylase (LSD1) activity [43]. The authors proposed a model in which a hyperglycemic stimulus increases the LSD1 demethylation of H3K9 near the *Mmp-9* promoter and subsequently increases *Mmp-9* expression [43]. Thus, the epigenome may represent a bridge between a hyperglycemic milieu that mediates DR pathogenesis through the regulation of gene expression for angiogenesis, endothelial-mesenchymal transition, and extracellular matrix remodeling.

### 2.4. Epigenetic Signals in Photoreceptor Degeneration and Retinal Dystrophies

Premature rod photoreceptor death and degeneration form the pathogenic basis for retinitis pigmentosa (RP) and related inherited retinal dystrophies (IRDs) [44]. Although these disorders are marked by extreme genetic heterogeneity, common epigenetic mechanisms in photoreceptor homeostasis could mediate downstream photoreceptor degeneration and represent a target of therapy. Rod photoreceptor genes such as *RHO* in humans undergo transcriptional regulation via DNA methylation at the promoter site [45]. The expression of *Rho* and other opsin photoreceptor genes is similarly regulated by DNA methylation in mice [46]. *rd1*-knockout mice experience spontaneous photoreceptor death and serve as a mouse RP/IRD model system. *rd1* mouse retinas showed increased H3K27me3 labeling during photoreceptor degeneration, driven by the histone lysine methyltransferase Ezh2 [15]. Intravitreal injection of an Ezh2 inhibitor targeting the catalytic SET domain was protective against cell death in the retina and decreased H3K27me3 staining [15]. In the *rd1* mouse RP model, rod photoreceptor death was marked by increased DNA methylation and DNA methyltransferase activity [47,48]. The authors analyzed retinas of wild-type (*wt*) versus *rd1* mice and found that in regions of the genome that were hypermethylated, gene ontology analysis revealed genes involved in nervous system development and function, as well as cell death and survival [47]. *rd1* retina explants treated with the DNA methyltransferase inhibitor decitabine showed decreased DNA methylation but had no effect on survival [47].

Dvoriantchikova and colleagues used data from human fetal retinas to investigate molecular epigenetic changes in genes associated with RP and related IRDs [49]. They found that photoreceptor-critical genes were hypermethylated in the RPC stage and underwent demethylation during differentiation to photoreceptors [49]. The authors argued that enzymes from the ten-eleven translocation (TET) family of DNA demethylases contribute to the demethylation of genes, which undergo demethylation in the RPC-photoreceptor transition based on *tet* knockout model systems, demonstrating impaired photoreceptor and RGC maturation [49]. Thus, they proposed a model of imbalance between DNA methylation and demethylation driven by mutations in photoreceptor-critical genes that may contribute to photoreceptor degeneration in RP and related IRDs [49].

In another study of epigenetic therapeutics in a mouse RP model, the authors studied *rd10* mice, in which rod photoreceptors undergo premature degeneration. An HDAC inhibitor and a lysine demethylase inhibitor were both protective against rod photoreceptor cell death [16]. Treatment with the lysine demethylase inhibitor also caused the upregulation of rod-specific genes including *Rho*, *Nr2e3*, and *Nrl* [16]. Interestingly, the lysine demethylase inhibitor and HDAC inhibitor decreased gliosis and microglial staining in the retina, implicating that a component of anti-inflammatory activity may involve the preservation of rod survival [16]. Another study of *rd10* mice found that the inhibition of bromodomain and extraterminal domain (BET) histone acetylation readers decreased photoreceptor degeneration and suppressed microglial activation in the retina [50].

Conversely, H3K27me3 has also been shown to protect photoreceptors from degeneration. In the *Pde6c^cpfl1^* mouse model of achromatopsia, cones undergo premature degeneration. Miller et al. investigated epigenetic inhibitors of cone survival in *Pde6c^cpfl1^* mice and found that an HDAC inhibitor was protective, enabling cone survival [51]. Histone demethylase inhibitor treatment affected global transcription in cones but did not have any effects on cell survival or electrophysiologic function in vivo, which the authors attributed partially to the single intravitreal treatment used [51]. When ex vivo retinal explants from *Pde6c^cpfl1^* mice were treated continuously with a histone demethylase inhibitor, improved cone survival and increased H3K27me3 staining were observed [51]. Thus, H3K27me3 signaling may serve distinct functions in rods versus cone photoreceptors.

## 3. Conclusions and Future Directions

### 3.1. Conclusions

The epigenome encompasses a complex array of molecular machinery tasked with the dynamic regulation of chromatin states and transcription in response to environmental and intercellular cues. Emerging evidence suggests that the epigenome may be a mediator of aging in the retina and throughout the body. Interestingly, a recent study from David Sinclair’s group proposed a mechanism for aging mediated by the repair of DNA double-stranded breaks, leading to the loss of the cell-specific epigenetic code, though it remains unknown whether this could drive pathogenesis in age-related eye conditions [52]. The development of the mammalian retina proceeds according to carefully orchestrated epigenetic regulation of retinal gene expression in RPCs, including DNA methylation, histone methylation, and histone acetylation. Age-related epigenetic changes in the retina, including the accumulation of DNA methylation, may represent the earliest responses to injury in diseases such as glaucoma, and the reversal of these epigenetic marks may represent a therapeutic target for age-related retinal disease [10]. In DR, the HDAC *Sirt6* serves as an environmental sensor of hyperglycemia, and its downregulation results in increased VEGF expression, which may represent an early step in proliferative DR pathogenesis [13]. The epigenome may also serve as an environmental integrator of local hypoxic and cytokine signaling in AMD and CNV pathogenesis [41]. Photoreceptor degeneration in RP also shows the dysregulation of epigenetic marks, but intriguingly, epigenetic drugs that inhibit histone methyltransferase Ezh2 have been shown to protect against rod degeneration in mouse RP models [15].

### 3.2. Current Trials in Genetic and Epigenetic Treatments

For retinal diseases characterized by the aberrant epigenetic regulation of gene expression, the epigenome represents an attractive therapeutic target. Many of the drugs targeting the epigenome are existing FDA-approved medications with extensive safety profiles and which have known epigenetic effects. For example, the anti-epileptic drug valproic acid/sodium valproate has HDAC inhibitory activity and has demonstrated neuroprotective effects in a rat model of Parkinson disease [4]. Valproic acid may have also conferred neuroprotection on RGCs in a rat ischemia/reperfusion model, as well as protecting against apoptosis in an optic nerve crush model [53]. Trichostatin A is another HDAC inhibitor which may also be protective against RGC apoptosis and stimulate RGC axonal regeneration [53]. Evidence from preclinical models of various HDAC inhibitors shows that they may exert neuroprotective effects by suppressing caspase-mediated apoptosis, downregulating TNF-α-mediated inflammation, and upregulating brain-derived neurotrophic factor (BDNF) [53]. HDAC inhibition may also reduce CNV lesion size and decrease VEGF levels in neovascular retinal conditions, potentially with similar benefits to invasive anti-VEGF intravitreal injections [38]. Thus, in preclinical models, epigenetic treatments, including HDAC inhibitors, show promise for degenerative, ischemic, and neovascular retinal conditions. Furthermore, epigenetic therapies may circumvent the cost, off-target effects, and delivery challenges associated with gene therapies, many of which require a viral vector and complex packaging and administration strategies. Epigenetic drugs may also show promise in the treatment of multifactorial retinal diseases such as AMD and DR, which have a complex environmental and genetic basis. They offer the advantage of systemic dosing, although the side effects of these drugs on other body organ systems have not been well characterized. To date, however, no epigenome-targeting drugs have advanced to clinical trials for retinal conditions, and additional efficacy and safety data are needed.

## Figures and Tables

**Table 1 genes-14-00417-t001:** Seminal findings from selected publications of epigenetic advances in retinal aging and disease.

Publication	Model System	Experimental Manipulation	Major Results	Significance of Findings
Lu et al. *Nature* 2020 [10] PMID 33268865	Mouse optic nerve crush and increased intraocular pressure glaucoma model	AAV2-packaged *Klf4*, *Oct4*, *Sox2* (OSK) injection	OSK injection accelerates optic nerve axon regeneration, decreases DNA methylation age OSK improves visual function/decreases DNA methylation age in OSK-treated mouse eyes following IOP insult	Pluripotency-associated gene expression rewinds the epigenetic clock and protects against optic nerve injury and may restore aspects of visual function
Hoshino et al. *Sci Rep* 2019. [11] PMID 30842553	Human fetal retina and cultured fetal retinal explants, pluripotent stem-cell-derived retinal organoids	RNA-Seq, DNA methylation analysis	DNA methylation age of the fetal retina correlates with chronologic age The foveal epigenetic clock is more advanced than the peripheral retinaRetinal organoid gene expression recapitulates the fetal retina	Validation of epigenetic clock in human retina Foveal development accelerated versus the peripheral retina Advanced epigenetic age in trisomy 21 retina
Xu et al. *Aging Cell* 2022 [12] PMID 36397653	Young and old mouse retina, IOP elevation	RNA-Seq, ATAC-Seq, H3K27 ChIP-Seq	Aging associated with changes in chromatin accessibility in retina, increased sensitivity to IOP elevation DNA methylation clock accelerated by IOP elevation	Disease states such as glaucoma may represent accelerated forms of molecular aging (i.e., senescence) DNA methylation may represent a target of therapy for senescence-associated diseases
Zorrilla-Zubilete et al. *J Neurochem* 2018 [13] PMID 29049850	Non-obese diabetic (NOD) mice, experimental hyperglycemia	*Sirt6* conditional knockout (KO), cultured Müller glia	Increased VEGF in NOD retina and in *Sirt6* KO *Sirt6* KO: increased H3K56/H3K9Ac levels *Sirt6* overexpressing Müller glia: decreased VEGF in response to hyperglycemia	SIRT6 acts as histone deacetylase in mouse retina, may act as molecular sensor in response to hyperglycemic in regulating *Vegf* expression
He et al. *Am J Physiol Endocrinol Metab* 2021. [14] PMID 33284093	Rat streptozotocin-induced DR model, cultured endothelial cells	Transfection of *Meg3* lncRNA	DR rat retina and hyperglycemia suppress *Meg3* expression, endothelial-mesenchymal transitionDNA methylation of *Meg3* promoter suppresses expression, promotes endothelial-mesenchymal transition	DR associated with endothelial-mesenchymal transition DNA methytransferase 1 acts as molecular sensor of hyperglycemia, causes methylation of long-noncoding RNAs Overexpression of lncRNA *Meg3* may represent a therapeutic strategy for DR
Mbefo et al. *AInt J Mol Sci* 2021 [15] PMID 34502238	Mouse *rd1* RP model	Ezh2 (histone methyltransferase) inhibitor	*rd1* photoreceptor degeneration associated with increased H3K27me3 Ezh2 inhibition slows photoreceptor death, decreases H3K27me3	Photoreceptor cell apoptosis in RP models may be mediated by increased H3K27me3, which could be delayed or prevented with histone methyltransferase inhibitor
Popova et al. *J Neurosci* 2021 [16] PMID 34193554	Mouse *rd10* RP model	Lysine demethylase (LSD1) inhibitor, HDAC inhibitor	HDAC and LSD1 inhibitors decrease rod degeneration, increase expression of rod genes Decreased gliosis and activated microglia in *rd10* retina treated with LSD1 or HDAC inhibitors	Rod photoreceptor degeneration may be mediated by histone lysine demethylation and histone deacetylation Treatment with HDAC or LSD1 inhibitor may slow rod photoreceptor degeneration

## Data Availability

Not applicable.

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
