# Peer review of "Advances in Ophthalmic Epigenetics and Implications for Epigenetic Therapies: A Review"

_genes, 2023, doi:10.3390/genes14020417_

Round 1
Reviewer 1 Report
The current work has summurized recent advances and discoveries in the field of epigenetics concerning the retina in a well-organized manner. Shedding the light to an important subject such as the epigenetic clock and the chronobiological age in relation to the epigenetical aging is an important aspect in understanding the most common age-related retinal diseases. The authors had covered different retinal pathologies starting from glaucoma to common inherited retinopathies such as retinitis pigmentosa. Lastely, authors highlighted the current trials in epigenetic treatments and although it is an attractive area for clinical advances, a need for more experimental studies is crucial.
Author Response
We appreciate the reviewer's kind words and time and effort in reviewing our manuscript.
Reviewer 2 Report
The manuscript by Moore et al. reviewed the recent advances in understanding the role of epigenetics in retinal development and disease. They focused on discussing epigenetic regulation of retinal development and epigenetic basis of aging and retinal diseases. The review covered most of the recent findings, and is of interest to readers in the field of ophthalmology. The authors also provided critical review of the field and proposed possible future directions.
Minor concern: It would be more clear if the authors can discuss individual retinal disease separately. For example, AMD and DR (also aging and RGC-involved diseases) can be discussed separately.
Author Response
We appreciate the reviewer's kind words and time and effort reviewing our manuscript and the opportunity to revise our submission.
1) We have separated the discussion of neovascular retinal conditions (2.3) into individual subheadings for Epigenetic basis of age-related macular degeneration (2.3.1), Epigenetic regulation of VEGF expression (2.3.2), and Epigenetic signals in hyperglycemic and diabetic retinopathy (2.3.3).
2) Additional background on glaucoma has been added in section 2.2.1: "Glaucoma encompasses a family of optic nerve disorders marked by progressive irreversible degeneration of RGCs which constitute the axons of the optic nerves. Age, family history, and intraocular pressure are strong risk factors, and glaucomatous optic neuropathies have been frequently used as experimental models for age-related neurodegeneration. "
3) Additional background on diabetic retinopathy added at beginning of section 2.3.3: "Diabetic retinopathy (DR) represents a common microvascular retinal complication of diabetes mellitus. Prolonged hyperglycemia may result in proliferative DR, in which neovascularization and tractional fibrotic membranes may result in irreversible vision loss. "
4) Additional background regarding genetic heterogeneity has been added to section 2.4: "While these disorders are marked by extreme genetic heterogeneity, common epigenetic mechanisms in photoreceptor homeostasis could mediate downstream photoreceptor degeneration and represent a target of therapy. "